# Stamped production of single-crystal hexagonal boron nitride monolayers on various insulating substrates

Fankai Zeng[1,2,9], Ran Wang[1,2,9], Wenya Wei[1,2,9], Zuo Feng[3,4,9], Quanlin Guo[3,4], Yunlong Ren [1,2], Guoliang Cui[1,2], Dingxin Zou[5], Zhensheng Zhang[5], Song Liu [5], Kehai Liu[6], Ying Fu[6], Jinzong Kou[1,2,6], Li Wang[7], Xu Zhou [1,2], Zhilie Tang[1,2], Feng Ding [8], Dapeng Yu [5], Kaihui Liu [3,4,6] ✉ & Xiaozhi Xu [1,2] ✉

Controllable growth of two-dimensional (2D) single crystals on insulating substrates is the ultimate pursuit for realizing high-end applications in electronics and optoelectronics. However, for the most typical 2D insulator, hexagonal boron nitride (hBN), the production of a single-crystal monolayer on insulating substrates remains challenging. Here, we propose a methodology to realize the facile production of inch-sized single-crystal hBN monolayers on various insulating substrates by an atomic-scale stamp-like technique. The single-crystal Cu foils grown with hBN films can stick tightly (within 0.35 nm) to the insulating substrate at sub-melting temperature of Cu and extrude the hBN grown on the metallic surface onto the insulating substrate. Single-crystal hBN films can then be obtained by removing the Cu foil similar to the stamp process, regardless of the type or crystallinity of the insulating substrates. Our work will likely promote the manufacturing process of fully single-crystal 2D material-based devices and their applications.

Due to the smaller size, higher speed and superior electrical performance of two-dimensional (2D) materials, devices based on fully 2D materials may lead to new applications in the future semiconductor industry[1–3]. The basic unit of a device consists of conductors, semiconductors, and insulators. Ideally, all of these materials should be single crystals to maximize the ultimate performance and uniformity of the devices. Therefore, in the past decade, numerous efforts have been employed to prepare single-crystal 2D conductors, semiconductors and insulators, and great progress has been achieved[4–14].

The size of typical 2D conducting graphene, semiconducting transition metal dichalcogenides (TMDs) and insulating hexagonal boron nitride (hBN) has increased from the micron to wafer scale.

To match mature industrial technology, the preparation of single-crystal 2D materials directly on insulating substrates is the ultimate pursuit for their high-end applications, as the transfer process would inevitably introduce uncontrollable contamination and mechanical damage, which would greatly degrade the performance of the devices[15–17]. For the growth of TMDs, the substrate does not need to

[1]Guangdong Basic Research Center of Excellence for Structure and Fundamental Interactions of Matter, Guangdong Provincial Key Laboratory of Quantum Engineering and Quantum Materials, School of Physics, South China Normal University, Guangzhou 510006, China. [2]Guangdong-Hong Kong Joint Laboratory of Quantum Matter, Frontier Research Institute for Physics, South China Normal University, Guangzhou 510006, China. [3]State Key Laboratory for Mesoscopic Physics, Frontiers Science Center for Nano-optoelectronics, School of Physics, Peking University, Beijing 100871, China. [4]International Centre for Quantum Materials, Collaborative Innovation Centre of Quantum Matter, Peking University, Beijing 100871, China. [5]International Quantum Academy, Futian District, Shenzhen 518045, China. [6]Songshan Lake Materials Laboratory, Institute of Physics, Chinese Academy of Sciences, Dongguan 523808, China. [7]Beijing National Laboratory for Condensed Matter Physics, Institute of Physics, Chinese Academy of Sciences, Beijing 100190, China. [8]Faculty of Materials Science and Engineering/Institute of Technology for Carbon Neutrality, Shenzhen Institute of Advanced Technology, Chinese Academy of Sciences, Shenzhen 518055, China. [9]These authors contributed equally: Fankai Zeng, Ran Wang, Wenya Wei, Zuo Feng. ✉e-mail: khliu@pku.edu.cn; xiaozhixu@scnu.edu.cn

participate in the decomposition of the precursor and only acts as a deposition surface, thus most TMDs can be directly grown on insulating substrates, and 2-inch single-crystal monolayer TMDs have been produced on sapphire substrates[11,12]. For the growth of graphene and hBN, the decomposition of methane and ammonia borane both requires high energy and usually involves catalytic substrate such as Cu[18,19]. Unfortunately, most insulating substrates are not catalytically active, which makes the controllable growth of monolayer single-crystal graphene or hBN films on them very difficult[20–26]. In the past few years, despite extensive efforts, graphene single crystals have only recently been prepared on specific substrates by a mass transport process[27,28], and the growth of a single-crystal hBN monolayer on insulating substrates is still a great challenge. Moreover, although remarkable progress has been made in the clean transfer of graphene, the improvement of hBN transfer is quite limited, and the surface cleanliness and film intactness are absolutely not comparable to those of graphene[15–17]. To data, it is impractical to replace hBN grown on insulating substrates with transferred hBN. Therefore, the controllable growth of hBN single crystals on insulating substrates has become one obstacle that restricts the fully 2D material-based devices.

Here, taking advantage of the premelting and adhesion of the Cu foil to the underlying substrate at a temperature close to the melting point, we successfully tuned the distance between hBN grown on the back surface of the Cu foil and the insulating substrate to a scale that can produce a strong attractive force, thus achieving a single-crystal hBN monolayer on the insulating substrate after removing the Cu foil. As hBN is first grown on the back surface of the Cu foil and then pasted onto the insulating substrate, the orientation control and seamless stitching of the hBN monolayer is consistent with that grown on the Cu foil[7,8]. This allows for us to obtain high-quality hBN films regardless of the type of substrate, and the growth of hBN single crystals is successfully realized on single-crystal $SiO_2(001)$, $SrTiO_3(001)$, c-plane sapphire and amorphous fused silica substrates. This approach is also demonstrated to be applicable to the growth of single-crystal graphene on all different insulating substrates. The prepared hBN and

graphene samples are demonstrated to be of high uniformity and quality. Our work fills in gaps in growing 2D single-crystal monolayers on insulating substrates and will likely promote the manufacturing process of fully single-crystal 2D material-based devices and the consequent tempting applications.

## Results

### Tuning distance between hBN and Cu foil

The van der Waals (vdW) interaction in 2D materials are closely related to the interlayer distance. For hBN on the $SiO_2$ substrate, the vdW force almost disappears when the distance is above 0.8 nm. With decreasing interlayer distance ($0.33 < r < 0.8$ nm), the attractive force increases steadily and then decreases to zero quickly, which drives $hBN/SiO_2$ closer together until reaching an equilibrium distance of approximately 0.33 nm (Fig. 1a-b, the corresponding potential energy curve is shown in Supplementary Fig. 1a). Therefore, when we place an hBN fake on the $SiO_2$ substrate, the interaction is very weak as the distance is usually much larger than 0.8 nm. If there is a way to reduce this distance to below 0.8 nm, the hBN may stick tightly to the insulating substrate due to its strong attractive force. Inspired by the stamp process, we proposed a method to extrude this distance by the tight adhesion of the Cu foil to the underlying substrate when heated near melting (Fig. 1c and Supplementary Fig. 2). Usually, the distance between the Cu foil and $SiO_2$ is ~120 μm at room temperature (Fig. 1d). When increasing the temperature to ~1080 °C, the Cu foils soften, and the distance decreases to ~20 μm (Fig. 1e). When the temperature continues to rise near the melting point, the Cu foil will further soften and stick tightly to the substrate, which has been demonstrated and used to grow bilayer graphene with arbitrary twist angles[29]. Cross-sectional high-resolution transmission electron microscopic (HRTEM) characterizations show that the distance is ~3.04 Å, which is smaller than 0.8 nm (Fig. 1f).

### Stamp-like growth of hBN on $SiO_2$

Utilizing this evolution of the space between the Cu foil and $SiO_2$ from ~120 μm to the atomic scale, we can tune the distance between hBN

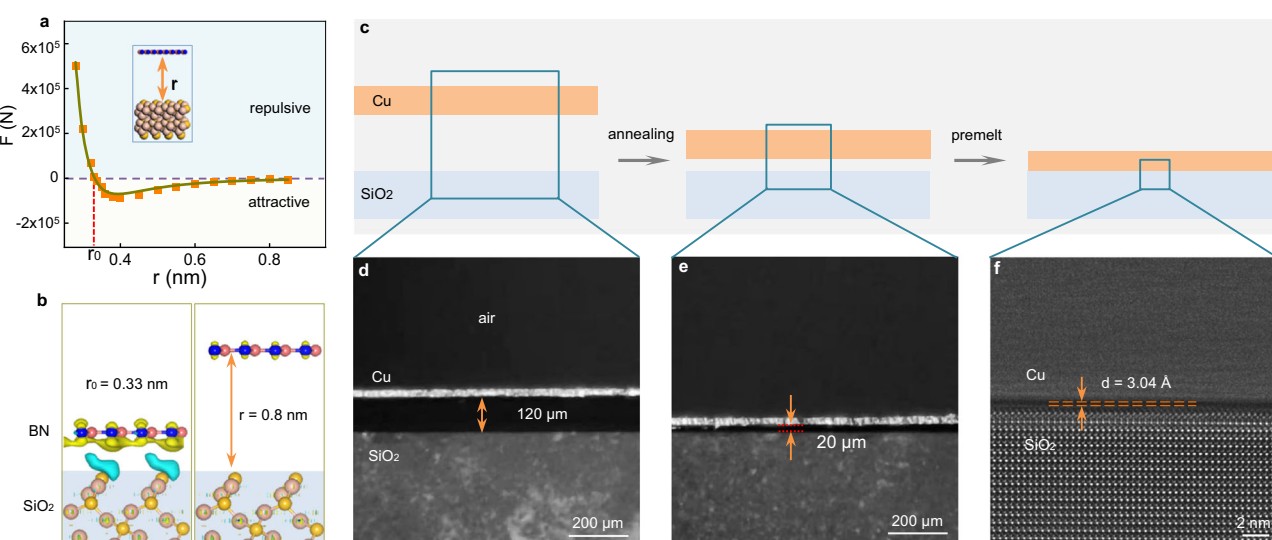

**Fig. 1 | Evolution of the space between Cu foil and the $SiO_2$ substrate with increasing temperature. a** Plot and fit of the van der Waals force as a function of the distance between hBN and $SiO_2$. The attractive force is negligible when the distance is larger than 0.9 nm and increases within the distance range of 0.33–0.8 nm. The orange squares represent the calculated data and the dark green line is the fitted curve. The light blue and light yellow shaded areas correspond to the repulsive and attractive force. Inset: schematic diagram of hBN on $SiO_2$.
**b** Charge density difference diagram of $hBN/SiO_2$ with distances of 0.8 nm and 0.33 nm. The yellow and blue colours correspond to the charge accumulation and

depletion, respectively. **c** Schematic diagrams of the evolution of the space between the Cu foil and $SiO_2$ substrate with increasing temperature. **d, e** Typical optical images of the cross section of the $Cu/SiO_2$ substrate annealed at room temperature (**d**) and 1080 °C (**e**). The cross-sectional optical images are obtained with a portable microscope at room temperature (see the setup in Supplementary Fig. 14). **f** Cross-sectional high resolution transmission electron microscopy image of the $Cu/SiO_2$ structure annealed at 1087 °C. The distance becomes smaller than 0.8 nm.

grown on the back surface of the Cu foil and the SiO$_2$ substrate. During hBN growth, the ~20 μm space allows the diffusion of the boron and nitrogen precursor and the single-crystal growth of hBN films on the back surface of vicinal Cu(110) by seamless stitching of unidirectionally aligned islands[7,8]. When the temperature further increases and the Cu/hBN/SiO$_2$ sandwich structure finally stabilizes, the strong attractive force is maintained between each interface (DFT calculations of the vdW forces between Cu/hBN and the SiO$_2$ substrate is shown in Supplementary Fig. 3), and allows the hBN film to remain intact during Cu removal (the detailed growth procedure can be seen in Methods, Fig. 2a, b). To check this "one-step growth and transfer" design, we first produced individual hBN islands on the SiO$_2$(001) substrate. Unidirectionally aligned hBN islands on the back surface of the Cu foil can be directly seen by optical microscopy through the SiO$_2$ as it is transparent (Fig. 2c, the adjusted figure is shown in Supplementary Fig. 4a to make the islands clearer) and the corresponding islands remained clean and intact after removing the Cu foil (Fig. 2d; for comparison, the optical image of the transferred hBN island is shown in Supplementary Fig. 5). Further second harmonic generation (SHG) mapping confirmed the successful growth of hBN islands and the seamless stitching of the merged islands (Fig. 2e).

Then, we produced inch-sized single-crystal hBN films by increasing the growth time, and verified the high quality, uniformity and cleanliness of the hBN monolayer prepared with our method through a series of characterizations (Fig. 3). The hBN film was very uniform without visible contaminants throughout the wafer (Fig. 3a, b and Supplementary Fig. 6). The single-crystal nature of hBN samples were verified with low energy electron diffraction (LEED) and selected area electron diffraction (SAED) characterizations in reciprocal space (Supplementary Figs. 7–8), and the high-resolution transmission electron microscopy (HRTEM) characterizations in real space (Supplementary Fig. 9). The $E_{2g}$ mode of hBN at 1367 cm$^{-1}$ and the full width at half maximum (FWHM) of approximately 13.6 cm$^{-1}$ in the Raman spectrum both confirmed the high-quality monolayer nature of the as-grown hBN sample[8]. Then the Raman spectra were further collected at a large scale, and the invisible peak offset and homogeneous intensity in the Raman map further verified its uniformity (Fig. 3d). X-ray

photoelectron spectroscopy (XPS) was also conducted to demonstrate the successful growth of hBN films on SiO$_2$(001) (Fig. 3e). The atomic ratio of B:N obtained from the XPS spectra is 1.08:1, which is very close to 1:1. No peaks corresponding to Cu and S can be detected, indicating no contaminations of Cu or the (NH$_4$)$_2$S$_2$O$_8$ etching solution (Supplementary Fig. 10). The surface cleanliness of the as-grown and transferred hBN samples was evaluated by atomic force microscopy (AFM), and the morphology images clearly showed that the surface of the as-grown hBN was much cleaner than that of the transferred sample (Fig. 3h).

## Universal growth of hBN and graphene

Therefore, with this simple stamp-like technique, the production of an inch-sized single-crystal hBN monolayer was realized. More importantly, as the hBN sample is first grown on the back surface of the Cu foil, our technique has two obvious advantages. (i) hBN grows directly on the back surface of the Cu foil, rather than at the interface by the very slow diffusion[27]. (ii) The single crystallinity and quality of the prepared hBN should be almost the same as those of the samples grown on Cu, as previously reported[7,8]. (iii) Since the only requirement of insulating substrates is heat resistance, the kinds of substrates are greatly expanded, and the universal production of hBN will work regardless of the composition and surface structure of the substrates, rather than the previous single-crystal c-plane sapphire[27,28]. To verify this universality, we grew hBN on single-crystal c-plane sapphire, SrTiO$_3$(001) and amorphous fused silica substrates successively, and all obtained high-quality intact films (Supplementary Fig. 11).

We then further validated the applicability of this approach to the growth of single-crystal graphene monolayers on various insulating substrates and the results once again proved the feasibility of our design. DFT calculations of the vdW force between graphene and sapphire varying with interlayer distance were conducted. Similar to hBN on the SiO$_2$ substrate, the vdW force almost disappears at distances larger than 1 nm, and is attractive within the distance range of 0.28–1 nm, which will drive graphene/SiO$_2$ together to an equilibrium distance of approximately 0.28 nm (Supplementary Fig. 12a–b). Using a similar stamp-like method, we grew individual graphene islands on

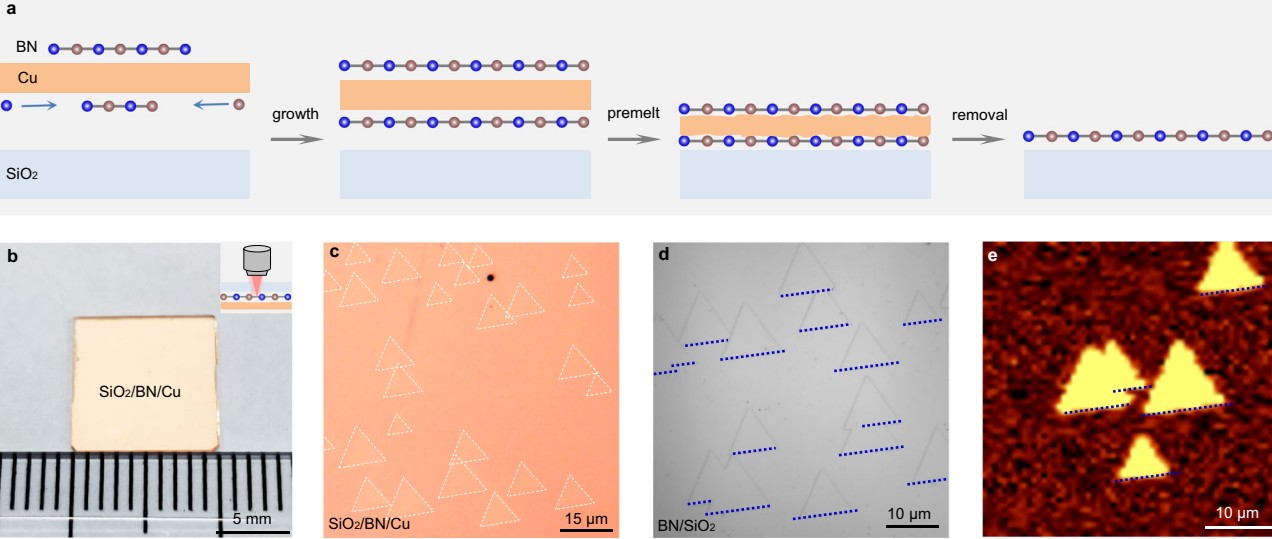

**Fig. 2 | Stamp-like growth of unidirectionally aligned hBN islands on SiO$_2$ substrates. a** Schematic diagrams of the growth process. The hBN islands are first grown on the back surface of the Cu foil and then tightly stick to the SiO$_2$ substrate. Finally, the Cu foil was removed to obtain hBN samples. **b** Optical image of the SiO$_2$/BN/Cu sandwiched structure. The image is taken from the back side of the SiO$_2$ substrate as it is transparent. Inset: schematic diagram of light transmission. **c** Zoom-in optical image of SiO$_2$/BN/Cu, unidirectionally aligned hBN islands can be

seen. The while dashed triangles represent the edges of hBN islands. The adjusted figure is shown in Supplementary Fig. 4a to make the islands clearer. **d** Optical image of as-grown hBN islands on SiO$_2$ after removing the Cu foil. The blue dashed lines represent the aligned edges of hBN islands. **e** Second harmonic generation intensity mapping of hBN islands. The homogeneous colour and absence of dark lines in the merged area confirm the intactness and seamless stitching of the sample. The blue dashed lines represent the aligned edges of hBN islands.

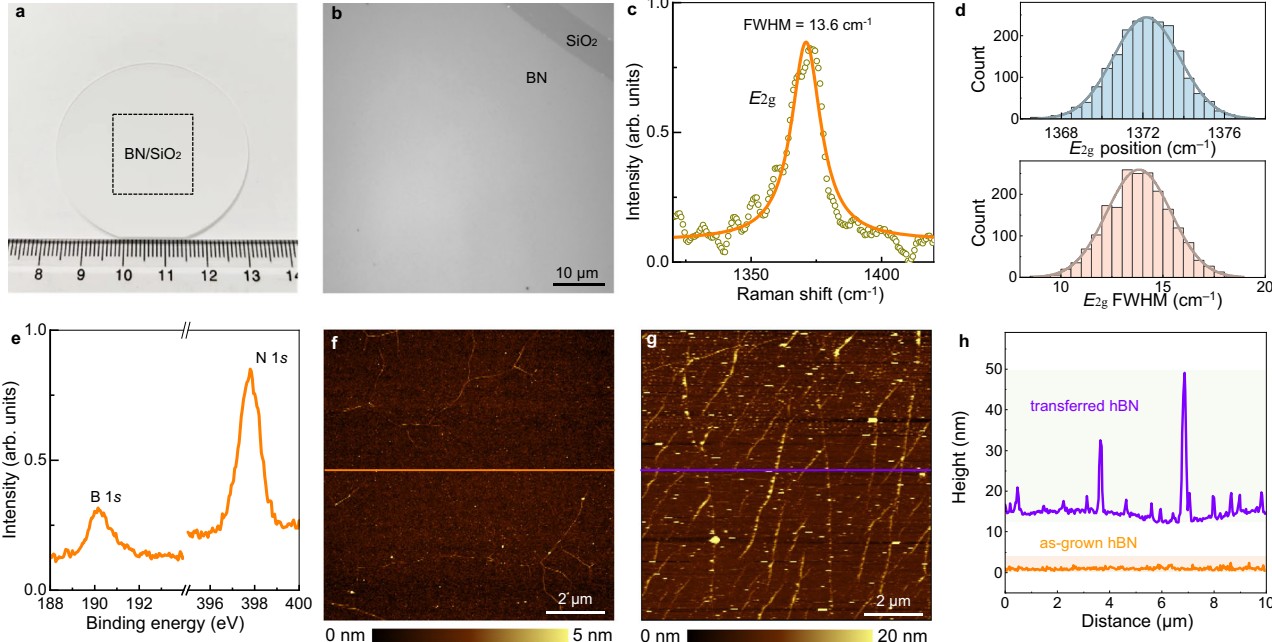

**Fig. 3 | Growth of single-crystal hBN monolayer films on SiO₂ substrates.**
**a** Optical image of an as-grown hBN sample. The black dashed squares represent the edges of the hBN films. **b** Zoom-in optical image of the as-grown hBN sample, which is clean and intact. The top right corner is an intentional scratch. **c** Raman spectrum of the hBN sample with a full width at half maximum (FWHM) of approximately 13.6 cm⁻¹ for the $E_{2g}$ mode, confirming the monolayer nature of hBN. The dark yellow circle represents the Raman data and the orange line is the fitted curve. **d** Statistical distributions of the $E_{2g}$-band position and FWHM of as-grown hBN. The light blue and light orange lines are the Gauss fitted curves. **e** XPS spectrum of the hBN sample. The characteristic peaks corresponding to B and N confirm the successful growth of hBN on the SiO₂ substrate. **f, g** AFM images of hBN films grown by the stamp-like method (**f**) and transfer (**g**), respectively. The orange and violet lines correspond to the positions of the height profiles in (**h**). **h** The height profile of (**f**) and (**g**). The grown hBN surface is much cleaner than the transferred surface. The light yellow and light orange shaded areas represent the hight distributions of transferred hBN and as-grown hBN.

the c-plane sapphire substrates and identified the surface cleanliness and intactness before and after Cu(111) removal (Supplementary Fig. 12c–d). Then, continuous single-crystal graphene films can be obtained and the statistical distributions of the 2D-band position and FWHM of graphene demonstrates the uniformity and monolayer nature (Supplementary Fig. 12e–f). Compared with the samples that directly grown on the quartz glass substrates using a widely studied catalyst-free method[30], graphene prepared by our technique shows higher quality, and the intensity ratio of the D/G mode is almost zero (Supplementary Fig. 12g). The strong interaction between graphene and the sapphire substrate was further verified in the Raman spectra. The 2D mode of graphene produced with our stamp-like method showed an obvious blueshift compared with the transferred graphene; and was even comparable to the sample grown on Cu(111) foils (Supplementary Fig. 12h), indicating strong strain produced during the cooling process[31,32]. Similarly, single-crystal SiO₂(001), SrTiO₃(001) and amorphous fused silica substrates were all tested and achieved the single-crystal growth of graphene films on all of them (Supplementary Fig. 13).

### Quality of hBN and graphene monolayers
To further check the quality of the synthesized graphene and hBN monolayers, we have carried out the electrical and dielectric measurements. For the transport property of graphene, eight field-effect transistor (FET) devices are directly fabricated and the obtained carrier mobility of as-grown graphene distribute very uniform, with an average value ~9100 cm²V⁻¹s⁻¹ at room temperature (Fig. 4a–c), which is comparable to that of graphene grown on Cu and much higher than that reported on insulators[33,34]. As comparison, the carrier mobility of transferred graphene varies over a wide range, from 1200 to 7600 cm²V⁻¹s⁻¹ (with an average value ~5100 cm²V⁻¹s⁻¹), due to the random contamination and cracks. For the dielectric characterizations, the hBN

samples grown on Cu and sapphire substrates were all transferred to the Au coated SiO₂/Si substrates and measured with conduction atomic force microscopy (CAFM). The typical breakdown current-voltage (I-V) curves show similar characteristics of capacitors with strongly non-linear onset of current (Fig. 4d, e), demonstrating that the quality of hBN samples grown on sapphire are comparable to that grown on Cu[35]. We also calculated the dielectric constant of hBN samples with scanning capacitance microscopy (SCM)[36], the average value is 3.62, which is consistent with previous theoretical and experimental data[37,38].

## Discussion
In summary, we have demonstrated the stamp-like technique to produce inch-sized single-crystal insulating hBN and conducting graphene monolayer films on various insulating substrates. Considering the fact that the controllable growth of 2-inch-sized single-crystal 2D semiconducting TMDs on sapphire substrates has been achieved recently[11,12], the direct production of 2D conductors, semiconductors and insulators on insulating substrates has been realized experimentally. Our results will push the manufacture and anticipated applications of fully single-crystal 2D material-based devices to take a solid step forward.

## Methods
### Production of single-crystal Cu (111) foil
A piece of commercially available poly-crystal Cu foil (25 μm thick, 99.8%; Sichuan Oriental Stars Trading Co. Ltd.) was first cleaned by electrochemically polishing with current intensity about 2 amperes for 30 seconds. The cleaned Cu foil was then placed on a flat fused silica and then loaded into a tube furnace (Tianjin Kaiheng, custom designed). The system was heated to 1080 °C in 1 h and maintained at this temperature for 2 h under 500 sccm Ar and 100 sccm H₂. After annealing, the system was cooled down naturally under 500 sccm Ar and 100 sccm H₂.

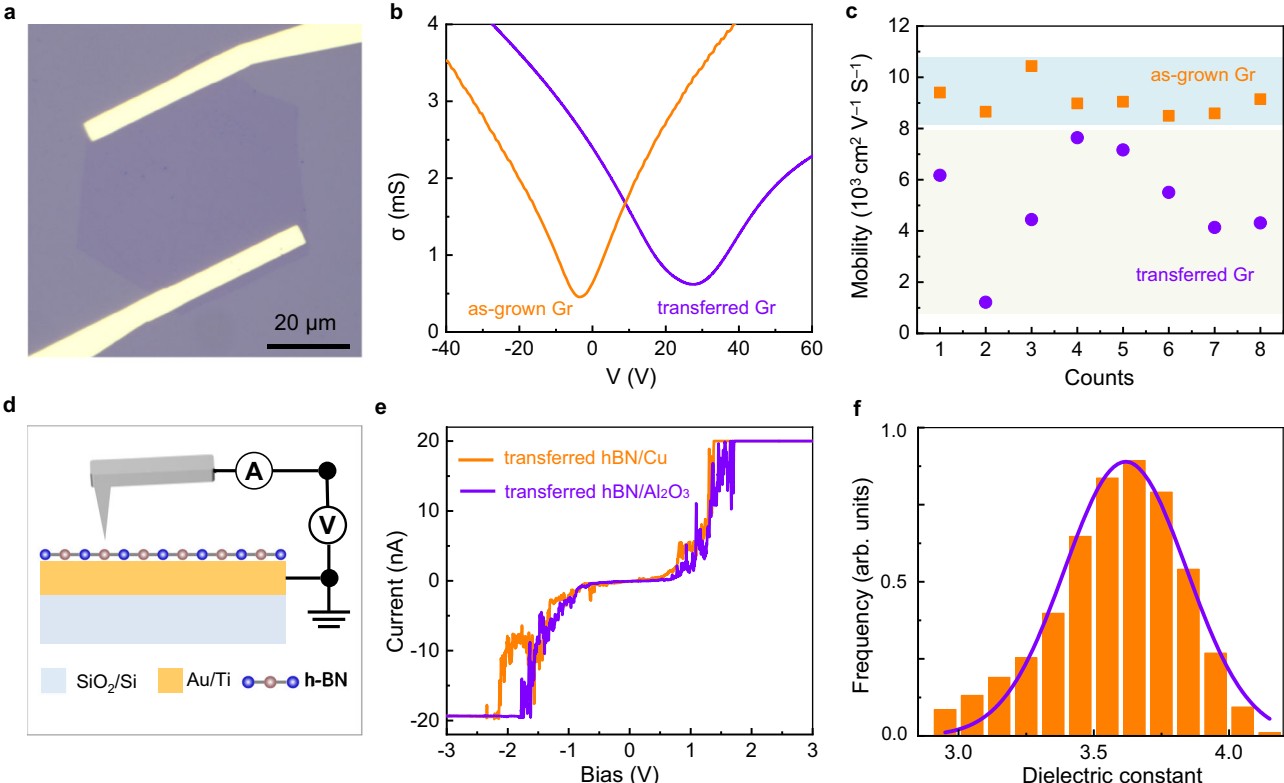

**Fig. 4 | Electrical and dielectric properties of graphene and hBN monolayers.**
**a** Typical optical image of the FET devices. **b** Plot of the conductivity of graphene as a function of gate voltage. The orange and violet curves correspond to the highest mobility of as grown graphene (~10,400 cm²/Vs) and transferred graphene (~7640 cm²/Vs), respectively. **c** Carrier mobility of as-grown graphene and transferred graphene on SiO₂/Si substrates. **d** Schematic diagram of the hBN samples and the electrical connections for the conduction atomic force microscopy. **e** Current-voltage characteristics of transferred hBN samples grown on Cu and Al₂O₃ substrates. **f** Distribution of calculated dielectric constant of the hBN samples with scanning capacitance microscopy. The violet solid line is the Gauss fitted curve.

## Production of single-crystal Cu (110) foil

A piece of commercially available poly-crystal Cu foil (25 μm thick, 99.8%; Sichuan Oriental Stars Trading Co. Ltd.) was first cleaned by electrochemically polishing with current intensity about 2 amperes for 30 s. A small piece of as-annealed single-crystal Cu (110) foil was placed on the surface of the cleaned Cu foil as a seed. Then they were placed on a flat fused silica and then loaded into a tube furnace (Tianjin Kaiheng, custom designed). The system was heated to 1085 °C in 1 h and maintained at this temperature for 5 min under 500 sccm Ar and 100 sccm H₂. Then the temperature was decreased to 1080 °C and annealed for 2 h. After annealing, the system was cooled down naturally under 500 sccm Ar and 100 sccm H₂.

## Growth of single-crystal hBN on various insulating substrates

A piece of the annealed Cu(110) was placed onto SiO₂(001)/SrTiO₃(001)/c-plane sapphire/fused silica substrates and loaded into a tube furnace (Tianjin Kaiheng, custom designed). The ammonia borane (roughly 0.85 mg, 97%; Aldrich) was placed into an aluminium oxide crucible and loaded into the upstream of the tube. The system was heated to 1080 °C in 1 h under 500 sccm Ar and 80 sccm H₂. Then, the flow of H₂ was switched to 20 sccm, and the aluminium oxide crucible was heated to 85 °C within 10 min using a heating belt and started the growth for 20 min/1 h to obtain individual hBN islands/continuous hBN film. After growth, the system was heated to 1087 °C in 4–8 min to make the Cu foil tightly stick to the substrate. Finally, the system was naturally cooled to room temperature with 500 sccm Ar and 100 sccm H₂.

## Growth of single-crystal graphene on various substrates

A piece of the annealed Cu(111) was placed onto SiO₂(001)/SrTiO₃(001)/c-plane sapphire/fused silica substrates and loaded into

a tube furnace (Tianjin Kaiheng, custom designed). Then, the system was heated to 1080 °C under 500 sccm Ar and 20 sccm H₂. During growth, 1 sccm of 1% CH₄ (diluted by Ar) was introduced as the carbon source for 30 min/1 h to obtain individual graphene islands/continuous graphene films. After the graphene growth on the back surface of the Cu foil, the furnace was heated to 1087 °C in 5–8 min to make the Cu foil tightly stick to the substrate. Finally, the system was naturally cooled to room temperature with 500 sccm Ar and 100 sccm H₂.

## Growth of graphene with traditional catalyst-free method

The quartz substrates were directly loaded into a tube furnace. Then, the CVD system was heated to 1080 °C in 1 h and annealed for 30 min with 500 sccm Ar. Then the growth was started with a gas mixture of 500 sccm Ar, 50 sccm H₂, and 30 sccm CH₄ for 4 h. Finally, the system was naturally cooled to room temperature with 500 sccm Ar and 50 sccm H₂.

## Removing Cu foils by chemical etching

The hBN/Cu/hBN/substrate (or the graphene/Cu/graphene/substrate) sandwiched sample was first cleaned with plasma for 2 min to etch the upper hBN (or graphene). Then they were immersed into (NH₄)₂S₂O₈ solution. After etching the Cu, the sample was immersed into deionized water for 10 min to remove residual solution.

## Production of graphene (hBN) films by directly peeling off Cu foils

The hBN/Cu/hBN/substrate (or the graphene/Cu/graphene/substrate) sandwiched sample can also be exfoliated by directly peeling off Cu foils and leave millimetre-scale hBN/substrate (or the graphene/

substrate) samples (see in Supplementary Fig. 15 for the procedure and the samples).

## Device fabrications and measurements

The FETs were fabricated through standard microfabrication process by electron beam lithography techniques on as-grown and transferred graphene on 300 nm $SiO_2$/Si. The Cr/Au contact electrodes (~5/50 nm) were fabricated by e-beam deposition system with a low vacuum ~$3 \times 10^{-7}$ Pa. All the electrical measurements were carried out in a probe station (base pressure $10^{-4}$ Pa) with Agilent semiconductor parameter analyzer (B1500, high resolution modules) at room temperature. The mobility is calculated by using the Equation $\mu = \frac{dI}{dVg} \cdot \frac{L}{W} \cdot \frac{1}{C_g V_{ds}}$. $W/L$ defines the channel size. The length of the channel $L$ is defined by the distance between the two electrodes. Considering the actual channel is larger than the electrode width, we calculate the width of the channel W by using equation $W = S/L$, where $S$ is the area of the graphene between the two electrodes.

**Characterization.** (i) AFM measurements were performed using Bruker Dimensional ICON under atmospheric environment. CAFM and SCM measurements were carried out using Cypher S under atmospheric environment. LEED measurements were performed using Omicron LEED systems in ultrahigh vacuum with a base pressure of <$3 \times 10^{-7}$ Pa. XPS measurements were conducted using Thermo Fisher ESCALAB XI+ system with a base pressure of <$5 \times 10^{-10}$ mbar.

(ii) Optical measurements. Optical images were conducted with an Mshot MSX10 microscope. Raman spectra were conducted on a WITec-Alpha300 Raman system with laser excitation wavelength of 532 nm and power of ~0.5 mW. SHG mapping was obtained using the same system under excitation from a femtosecond laser centred at 820 nm with average power of 800 µW (Spectra-Physics Insight system with pulse duration of 100 fs and repetition rate of 80 MHz).

(iii) TEM measures. The TEM cross-section samples were prepared by the focused ion beam (FIB) technique (ThermoFisher Helios G4 UX) with a gallium ion source operated at 30 kV and 2 kV. The STEM experiments were carried out on an FEI Titan Temis G2 300 system operated at 300 kV.

## Computational details

Geometric optimization and energy calculations of the hBN/$SiO_2$ systems and graphene/sapphire systems were carried out using density functional theory (DFT) as implemented in Vienna Ab-initio Simulation Package. The generalized gradient approximation (GGA) with the Perdew–Burke–Ernzerhof (PBE) exchange-correlation function was used with the plane-wave cutoff energy set at 400 eV for all calculations. The dispersion-corrected DFT-$D_3$ method was used because of its good description of long-range van der Waals interactions for multilayered 2D materials. The geometries of the structures were relaxed until the force on each atom was less than 0.02 eV Å$^{-1}$, and the energy convergence criterion of $1 \times 10^{-4}$ eV was met. In the out-plane direction, the vacuum spacing between neighbouring images is set at least 10 Å to avoid a periodic imaging interaction.

## Data availability

The Source Data underlying the figures of this study are available with the paper. All raw data generated during the current study are available from the corresponding authors upon request. Source data are provided with this paper.

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

## Acknowledgements

This work was supported by the National Key R&D Program of China (2022YFA1403500 (K.L.)), the National Natural Science Foundation of China (12322406 (X.X.), 52102043 (X.X.), 52025023 (K.L.), 51991342 (K.L.), 52021006 (K.L.), 52372046 (X.Z.) and 52102044 (X.Z.)), Guangdong Basic and Applied Basic Research Foundation (2020B1515020043 (X.X.), 2019A1515110302 (J.K.)), Guangdong Major Project of Basic and Applied Basic Research (2021B0301030002 (K.L.)), Science and Technology Program of Guangzhou (2019050001 (X.X.)), the Key R&D Program of Guangdong Province (2020B010189001 (X.X.), 2019B010931001 (K.L.), 2018B010109009 (D.Y.) and 2018B030327001 (D.Y.)), the Pearl River Talent Recruitment Program of Guangdong Province (2019ZT08C321 (X.X.)), the National Postdoctoral Program for Innovative Talents (BX20220117 (W.W.)), China Postdoctoral Science Foundation (2022M721224 (W.W.)), the Key Project of Science and Technology of Guangzhou (202201010383 (F.Z.)) and the Strategic Priority Research Program of Chinese Academy of Sciences (XDB33000000 (K.L.)). We thank the National Supercomputer Centre in Tianjin for computing support.

## Author contributions

X.X. and Kaihui Liu supervised the project. F.Z., R.W., G.C., J.K., L.W. and D.Z. conducted the sample growth, D.Y., Z.Z., S.L. and Q.G. performed the TEM experiments. Y.R., Y.F. and Kehai Liu performed the AFM experiments. Z.F. performed the electrical measurements. W.W. and F.D. performed the theoretical calculations. X.X., F.D. and Kaihui Liu wrote the article, Z.T. and X.Z. revised the manuscript. All of the authors discussed the results and comments on the paper.

## Competing interests

The authors declare no competing interests.
