## [Peer Review File · Nature Communications]

Stamped production of single-crystal hexagonal boron nitride monolayers on various insulating substratesEditorial Note: Parts of this Peer Review File have been redacted as indicated to remove third-party material where no permission to publish could be obtained.

REVIEWER COMMENTS

Reviewer #1 (Remarks to the Author):

In this manuscript, the authors propose a method for the growth of 2D materials on insulating substrates, with a focus on monolayer hBN. The hBN grown is performed on the surface of a Cu foil supported on an insulator surface. The hBN and the insulator are then brought into contact by heating at temperatures close to the melting point of Cu. This covers a need in the obtention of high quality 2D materials on the top of arbitrary substrates, without the constraints of using a catalytic surface. After a careful evaluation, I have some concerns about the results and the manuscript that should be addressed before I can recommend its publication.

- I would like to remark that to me the proposed work describes a "high-temperature transfer" rather than a direct "growth on an insulator", although I understand that this can be debatable. The hBN (graphene) grows on the Cu foil, and then adheres to the target substrate at high temperatures. Although the authors refer to this process as a "stamp-like method" (see "grown by the stamp-like method" in the caption of Fig. 3), it should be noted that "stamp" methods are typically used for transferring 2D materials rather than growing them. In my opinion, the present work should be considered more of an "automatic transfer" or "one-step growth and transfer" technique than a direct growth method on insulating substrates. I would like to reiterate that this is a personal point of view, and I would understand if it were not shared by the authors.
- The method proposed seems to be derived from another reported by some of the authors for the synthesis of large areas of twisted bilayer graphene (DOI: 10.1038/s41563-022-01361-8). In that mentioned work, monolayer graphene is grown on the surface of two different Cu foils. The bilayer graphene is then formed by heating to temperatures close to the Cu melting point. These similarities would need to be acknowledged in the current work.
- The authors claim that the hBN is single-crystal, but this assertion appears to be based solely on the SHG data in Fig. 2e. The figure shows a small amount of mostly isolated hBN

grains, so claiming the hBN with a full surface coverage to be single-crystal does not seem to be backed by evidence, even though the grains are initially aligned. It is also worth mentioning that this initial alignment is based on optical images, which do not seem accurate enough to justify the claims of single-crystal after merging. I understand that in previous literature (e.g., DOI: 10.1038/s41586-019-1226-z from the same authors) “perfectly aligned grains” seem to almost automatically imply “seamless stitching”. Such argument does not fully convince me, and I tend to think of the alignment as a necessary but not sufficient condition for the seamless stitching. So, I would like to ask the authors to justify the single-crystal claims and/or provide stronger experimental evidence. All the previously said also applies to claims of single-crystal graphene.

- Concerning the DFT calculations for the vdW forces between hBN (or graphene) and the insulator substrate, the presence of the Cu is never considered. Given the monolayer nature of the hBN and graphene in this work, is it a reasonable assumption neglecting the effect of the Cu?

- The distance between the hBN and the insulating substrate needs to be well below 1 nm to get a significant attractive force between them, according to the DFT calculations. However, there is no experimental evidence that the distances decreased below that range upon heating the Cu. The only evidence shown is the optical images 1d-f, but these do not allow measuring such extremely small gaps. Thus, the claim made in the manuscript of “the strong attractive force is maintained between each interface, and allows the hBN film to remain intact during Cu removal” is not really supported by experimental evidence.

Concerning this, there are for example works that demonstrate the possibility of transferring graphene to an insulator substrate by etching the Cu existing between them (as e.g., DOI: 10.1021/acsami.2c16505 and probably others): i.e., in principle there is no need of a strong interaction between the 2D material and the target substrate prior to the removal of the Cu, and the fact that the hBN remains in the insulating substrate after etching the Cu does not prove they were that close at the beginning.

- Related to the previous claim about the lack of experimental evidence on the actual distances between hBN and substrate, the claim of the distance becoming “invisible” when temperature increases to near the melting point seems like an improper and misleading term. This appears both in the text and in the caption of figure 1.

- In the Supplementary Fig. 6, the quality of graphene produced by the current method is

compared with some other graphene. However, it is not clear how this other graphene was synthesized and/or transferred, and the only mention is this being the “traditional method”. The authors should explain how that sample was prepared. They should also compare the results with those from graphene grown on a Cu foil and transferred to the same kind of substrate. As an example, the comparison of the D/G Raman intensity ratio of Sup. Fig. 6g seems very biased towards the current method. A more useful comparison would be with e.g., the results from reference DOI: 10.1016/j.scib.2017.07.005 by the same authors, in which the graphene is transferred following a widely adopted PMMA-mediated process.

- Related to the possible bias towards the proposed method, the conductivity plot of transferred graphene shown in Fig. 4b does not seem to be that good. It is fairly easy to find examples of better performing graphene devices in the literature prepared by similar method (PMMA-mediated transfer). What is the mobility for that specific device? Is it a representative device (i.e., with a mobility of around 5000 or 6000 cm²/Vs, according to Fig. 4c), or is it the worst performing one (with mobility around 1000, and clearly an outlier)? About the mobility, the authors should also indicate how it was calculated (mainly how the channel sizes were determined). It would also be good to see the position of the Dirac point, which in Fig. 4b it has been zeroed, to estimate how much doped is the graphene. This should be also used to compare between the different methods.

- Supplementary Figure 4 displays optical images of hBN on SiO₂. Notably, images c-h appear almost identical, with the same spots in the same positions. This does not seem to be caused by artifacts from the lens or equipment, as other images most probably taken with the same setup do not exhibit these spots (Sup. Fig. 4a,b, Fig. 3b of the main text,...). The authors should provide an explanation for this intriguing observation.

- The XPS spectrum only shows the B and N peaks, and there is no discussion about the results. What is the B:N ratio obtained from XPS? Also, does XPS show any evidence of contamination, such as Cu or residues from the etching solution?

- Si coming from the quartz tube at high temperatures can have an influence on the growth of graphene (see e.g., DOI: 10.1038/s41565-020-0743-0). In the current method, the growth of hBN (graphene) occurs very close to the SiO₂ surface (or to the Al₂O₃). Can contamination from the insulator substrate play a role in the growth of the 2D materials here? Is it completely excluded the possible contamination of the 2D material with material from the substrate?

- Usually, monolayer hBN on Cu presents a very low contrast, so it is a bit surprising that it is so clearly seen in the image of Fig. 2c. As the image is taken from the SiO₂ side, the contrast should not be enhanced by the oxidation of the Cu. Is there any reason for the hBN being that visible?
- Small mistakes: “height profile” instead of “height distribution” in caption of Fig. 3; “garphene” in caption of Sup. Fig. 6.

Reviewer #2 (Remarks to the Author):

The manuscript by Zeng et al. reports a stamp-like process that can extrude hBN or graphene from metallic substrates and tightly stick on various insulating substrate. This method demonstrates an “in-situ transfer” process that can transfer hBN or graphene that were grown initially on a Cu foil without using any postgrowth transfer process.

Characterizations show that the hBN and graphene are flat and clean with rather good electrical and dielectric performance. The method demonstrated here is very unique and potentially useful for solving the long-standing problems caused by the inevitable postgrowth transfer processes. Therefore, I tend to support publication of this work, after addressing my comments:

1. The title of “stamped growth” is quite confusing. This method is actually a stamp-like transfer process after growth, not exactly a growing process. I suggest the authors to change this term.
2. In Fig. 1d-f, how did the authors take the cross-sectional optical images? I am curious about how to take such image at 1080 and 1087 oC. Please explain.
3. Does this method work as well by using a two-step process in two separate furnaces? (i.e, growing 2D materials on a metallic substrate in a furnace and performing the stamp-like transfer in another furnace.) In principle, it should work even in two separate furnaces. If not, do the authors have any comment on this.
4. The stamp-like process has to be performed at the melting temperature of Cu. This is a very high temperature, which may damage the substrate with a very thin dielectric layer. Such a high temperature is also not suitable for back-end-of-line processes. Could the authors comment on this?

Reviewer #3 (Remarks to the Author):

The manuscript entitled "Stamped Growth of Single-Crystal Hexagonal Boron Nitride Monolayers on Various Insulating Substrates" introduces a method of interface growth between Cu foil and insulating substrates. The authors claim that the Cu foil and insulator can be joined together via a premelt process at around the melting point of the Cu foil. After the premelt process, they grew a single-crystal hBN film between the Cu foil and SiO₂, resulting in hBN/Cu/hBN/insulating substrates. The upper hBN layer was etched using plasma treatment, and then the Cu/hBN/insulating substrates were immersed in an (NH₄)₂S₂O₈ solution to remove the Cu, resulting in hBN/insulating substrates. However, the method and results presented in this manuscript lack novelty compared to previous reported research. Additionally, there are several sections of the manuscript that could lead to confusion or misunderstanding. Therefore, this manuscript should be rejected from Nature Communications. I left some representative comments as below.

1. Many previous studies have achieved monolayer single-crystalline hBN films. In this work, the authors only applied interface growth, and although they demonstrated a chemical etching process to remove the upper hBN and Cu, they also transferred the prepared hBN/SiO₂ to another substrate. Therefore, the novelty of this work compared to previous reports is unclear.
2. The concept of this work is similar to a previous report [Nature Communications 11, 849 (2020)]. In that report, the metal substrate could be easily removed after growth. In contrast, this work applied chemical etching and transfer methods.
3. If this work can demonstrate interface growth without the need for transfer processes and is applicable to universal substrates such as PET film or glass, it would have strong novelty.
4. Finally, the manuscript lacks clear explanations about the preparation of samples, measurements, and analysis. Therefore, all specific details should be included in the manuscript.

Reply to Referee #1

Original comment (1):

In this manuscript, the authors propose a method for the growth of 2D materials on insulating substrates, with a focus on monolayer hBN. The hBN grown is performed on the surface of a Cu foil supported on an insulator surface. The hBN and the insulator are then brought into contact by heating at temperatures close to the melting point of Cu. This covers a need in the obtaintion of high quality 2D materials on the top of arbitrary substrates, without the constraints of using a catalytic surface. After a careful evaluation, I have some concerns about the results and the manuscript that should be addressed before I can recommend its publication.

Our reply:

We thank the referee's efforts and time in reviewing our manuscript. His/her valuable suggestions and comments are quite helpful for us to improve the quality of this work. We have added more experimental and theoretical data to address these concerns as in the below replies.

Original comment (2):

1. I would like to remark that to me the proposed work describes a "high-temperature transfer" rather than a direct "growth on an insulator", although I understand that this can be debatable. The hBN (graphene) growths on the Cu foil, and then adheres to the target substrate at high temperatures. Although the authors refer to this process as a "stamp-like method" (see "grown by the stamp-like method" in the caption of Fig. 3), it should be noted that "stamp" methods are typically used for transferring 2D materials rather than growing them. In my opinion, the present work should be considered more of an "automatic transfer" or "one-step growth and transfer" technique than a direct growth method on insulating substrates. I would like to reiterate that this is a personal point of view, and I would understand if it were not shared by the authors.

Our reply:

We greatly thank the referee for the suggestion on the use of "stamp". We agree with the referee that our work is a "one-step growth and transfer" technique. Following the referee's advice, we have changed the title as "Stamped production of single-crystal hexagonal boron nitride monolayers on

various insulating substrates” in the revised manuscript, where the word “production” contains both the growth and transfer process in one step. And in the text part, we described our idea more clearly about the “one-step growth and transfer” design.

Original comment (3):

2. The method proposed seems to be derived from another reported by some of the authors for the synthesis of large areas of twisted bilayer graphene (DOI: 10.1038/s41563-022-01361-8). In that mentioned work, monolayer graphene is grown on the surface of two different Cu foils. The bilayer graphene is then formed by heating to temperatures close to the Cu melting point. These similarities would need to be acknowledged in the current work.

Our reply:

We thank the referee for this kind suggestion. We agree with the referee that our work and the work of twisted bilayer graphene both take advantage of the pre-melting of the Cu foils at high temperature. These two works may have relevance; however, the key experimental designs of the two works are fundamentally different. For the bilayer graphene work, as the couplings between Cu and graphene are larger than that between two graphene layers, the twist angle of bilayer graphene can be locked by two twisted Cu(111) foils, and this is the key point of the design. In this work, the Cu foils grown with hBN films can stick tightly (at atomic scale) to the insulating substrate at pre-melting temperature of Cu and then extrudes the hBN grown on the metallic surface onto the insulating substrate, and the key point is to tune the distance between hBN and the substrate to form strong vdW interaction.

Following the referee’s advice, we have added the above discussion in the revised manuscript.

Original comment (4):

3. The authors claim that the hBN is single-crystal, but this assertion appears to be based solely on the SHG data in Fig. 2e. The figure shows a small amount of mostly isolated hBN grains, so claiming the hBN with a full surface coverage to be single-crystal does not seem to be backed by evidence, even though the grains are initially aligned. It is also worth mentioning that this initial alignment is based on optical images, which do not seem accurate enough to justify the claims of

single-crystal after merging. I understand that in previous literature (e.g., DOI: 10.1038/s41586-019-1226-z from the same authors) “perfectly aligned grains” seem to almost automatically imply “seamless stitching”. Such argument does not fully convince me, and I tend to think of the alignment as a necessary but not sufficient condition for the seamless stitching. So, I would like to ask the authors to justify the single-crystal claims and/or provide stronger experimental evidence. All the previously said also applies to claims of single-crystal graphene.

Our reply:

We sincerely thank the referee for raising this important concern on the experimental evidence of the single crystallinity. Following the referee’s advice, we have done more experiments to demonstrate the aligned orientation and seamless stitching of both graphene and hBN samples. For the aligned orientation, we have conducted low energy electron diffraction (LEED) characterizations with typical spot size of 100 μm , the identical orientations of the diffraction patterns in different areas proved that graphene and hBN samples all have the same lattice orientation (Fig. R1).

For the seamless stitching, we have performed H_2 etching, the uniform hexagonal and triangular holes without defective lines proved that there are no grain boundaries (Fig. R2). These isolated holes are from the point defects instead of the grain boundaries.

We have added these data and discussions in the revised manuscript.

Fig. R1 LEED patterns of graphene (a-f) and hBN (g-l) samples at different areas. The identical orientations confirm the single-crystal nature.

Fig. R2 Optical images of graphene (a-b) and hBN (c-d) samples after H₂ etching at different areas.

Original comment (5):

4. Concerning the DFT calculations for the vdW forces between hBN (or graphene) and the insulator substrate, the presence of the Cu is never considered. Given the monolayer nature of the hBN and graphene in this work, is it a reasonable assumption neglecting the effect of the Cu?

Our reply:

We thank the referee for this kind suggestion and totally agree with the referee that the presence of Cu should be considered. Following the referee's advice, we have conducted additional DFT calculations of the vdW forces between hBN (graphene) and the insulating substrate considering the presence of Cu (Fig. R3a-b). Similar to the result in the last version of the manuscript, for hBN, the vdW force after considering Cu almost disappears at distances larger than 0.8 nm (1 nm for graphene), and is attractive within the distance range of 0.33-0.8 nm (0.29-1 nm for graphene), which will drive Cu/hBN and the substrate together to reach an equilibrium distance of ~0.33 nm

(~0.29 nm for graphene, Fig. R3c-d).

We have updated this data in the revised manuscript.

Fig. R3. Schematic diagrams of the Cu/hBN/SiO₂ (a) and Cu/graphene/Al₂O₃ structure (b). Plots and fit of the van der Waals force as a function of the distance between Cu/hBN/SiO₂ (c) and Cu/graphene/Al₂O₃ (d). For hBN, the attractive force is negligible when the distance is larger than 0.8 nm and increases with the decreasing of distance at the range of 0.33-0.8 nm. For hBN, the attractive force is negligible when the distance is larger than 1 nm and increases with the decreasing of distance at the range of 0.29-1 nm.

Original comment (6):

5. The distance between the hBN and the insulating substrate needs to be well below 1 nm to get a significant attractive force between them, according to the DFT calculations. However, there is no experimental evidence that the distances decreased below that range upon heating the Cu. The only evidence shown is the optical images 1d-f, but these do not allow measuring such extremely small gaps. Thus, the claim made in the manuscript of “the strong attractive force is maintained between each interface, and allows the hBN film to remain intact during Cu removal” is not really supported by experimental evidence. Concerning this, there are for example works that demonstrate the possibility of transferring graphene to an insulator substrate by etching the Cu existing between them (as e.g., DOI: 10.1021/acsami.2c16505 and probably others): i.e., in principle there is no need of a strong interaction between the 2D material and the target substrate prior to the removal of the Cu, and the fact that the hBN remains in the insulating substrate after etching the Cu does not prove they were that close at the beginning.

Our reply:

We greatly thank the referee for pointing out this important concern. We agree with the referee that the atomic-scale characterization of the interface is necessary and the optical image shown in Fig. 1d-f is not direct information on the interfacial distance. Following the referee’s advice, we have carried out cross-sectional high-resolution transmission electron microscopic (HRTEM) characterization. The atomic characterizations show the distance of Cu and SiO₂ after pre-melting is 3.04 Å (Fig. R4), which is consistent with our theoretical calculations. We have updated this data as Fig. 1f in the revised manuscript.

Fig. R4 Cross-sectional HRTEM image of the Cu/SiO₂ structure.

We also thank the referee to provide the related reference. In that work, the transfer of graphene was realized by their penetration etching method (ACS Appl. Electron. Mater. 2020, 2, 238). In the previous method, the etchant can etch the Cu through penetration of the organic supporting PMMA layer and graphene. Then, the PMMA/graphene can be obtained on SiO₂ substrate (Fig. R5a). This work is quite useful but still has some shortage. For example, the method needs the PMMA as a supporting layer, otherwise the film will be broken during the etching of the Cu.

In our method, we do not want to involve any organic supporting layer (Fig. R5b). In this case, if there is no strong interaction between hBN (graphene) and the substrate (or the distance of hBN/graphene and the substrate is not small enough), the hBN (graphene) will be broken in the process of etching the Cu foil. So, the small enough distance and the strong enough interaction is critical in our method.

Fig. R5 Schematic diagram of the etching process of Cu with (a) and without (b) PMMA supporting layer.

Original comment (7):

6. Related to the previous claim about the lack of experimental evidence on the actual distances between hBN and substrate, the claim of the distance becoming “invisible” when temperature increases to near the melting point seems like an improper and misleading term. This appears both in the text and in the caption of figure 1.

Our reply:

We greatly thank the referee for pointing out this inaccurate description of “invisible”. We have added the HRTEM images (Fig. R4) and updated text and the caption of Fig. 1f in the revised manuscript.

Original comment (8):

7. In the Supplementary Fig. 6, the quality of graphene produced by the current method is compared with some other graphene. However, it is not clear how this other graphene was synthesized and/or transferred, and the only mention is this being the “traditional method”. The authors should explain how that sample was prepared. They should also compare the results with those from graphene grown on a Cu foil and transferred to the same kind of substrate. As an example, the comparison of the D/G Raman intensity ratio of Sup. Fig. 6g seems very biased towards the current method. A more useful comparison would be with e.g., the results from reference DOI: 10.1016/j.scib.2017.07.005 by the same authors, in which the graphene is transferred following a widely adopted PMMA-mediated process.

Our reply:

We thank the referee for the suggestion on the comparison of graphene produced in different methods. The graphene prepared with “traditional method” is the sample that directly grown on the quartz glass substrates using a widely studied catalyst-free APCVD method (Advanced Materials 2019, 31, 1803639; Nano Letters 2015, 15, 5846; Advanced Materials 2015, 27, 7839; Nano Research 2015, 8, 3496). The quartz substrates were directly loaded into a tube furnace. Then, the

CVD system was heated to 1080 °C in 1 hour and annealed for 30 min with 500 sccm Ar. Then the growth was started with a gas mixture of 500 sccm Ar, 50 sccm H₂, and 30 sccm CH₄ for 4 hours. We have added the growth procedure in the revised Method part and described how the control sample was prepared.

We also compared the samples grown on Al₂O₃ using our method with that grown on a Cu foil and transferred to Al₂O₃ substrate in Supplementary Fig. 6h. The 2D mode of graphene produced with our stamp-like method showed an obvious blue shift compared with the transferred graphene, indicating strong interaction between graphene and the sapphire substrate. Due to the mature transfer of graphene, we did not compare the D/G Raman intensity ratio of the samples obtained with our stamp-like method and the one transferred following a PMMA-mediated process (Fig. R6). At present, the D band of transferred graphene samples is also negligible. So, their quality were further tested through the electrical characterizations.

It is worthwhile to note that although remarkable progress has been made in the clean transfer of graphene, the progress of hBN transfer is quite limited, and the surface cleanliness and film intactness are absolutely not comparable to those of graphene (Nature Communications 2019, 10, 1912; Nature Communications 8, 14560; Science Advances 2015, 1, e1500222).

Fig. R6 Typical Raman spectra of graphene samples that grown on Cu (violet curve), grown on Al₂O₃ (dark green curve), and transferred to Al₂O₃ (orange curve).

Original comment (9):

8. Related to the possible bias towards the proposed method, the conductivity plot of transferred graphene shown in Fig. 4b does not seem to be that good. It is fairly easy to find examples of better performing graphene devices in the literature prepared by similar method (PMMA-mediated

transfer). What is the mobility for that specific device? Is it a representative device (i.e., with a mobility of around 5000 or 6000 cm²/Vs, according to Fig. 4c), or is it the worst performing one (with mobility around 1000, and clearly an outlier)? About the mobility, the authors should also indicate how it was calculated (mainly how the channel sizes were determined). It would also be good to see the position of the Dirac point, which in Fig. 4b it has been zeroed, to estimate how much doped is the graphene. This should be also used to compare between the different methods.

Our reply:

We greatly thank the referee for raising the concern on the mobility of graphene. Indeed, the mobility of transferred graphene reported in the literature can be very high by involving the hBN. But the mobility obtained directly on SiO₂/Si substrate at room temperature is not so high (Nature Electronics 2023, 6, 126; Nature Communications 2022, 13, 5410; Nature Materials 2022, 21, 740; Nature 2021, 596, 519; Science Advances 2021, 7, eabk0115). Also, we note that the obtained mobility values of CVD graphene are highly related to the transfer processes in different groups. So it is more suitable to compare the value in the same group. In our previous work, the mobility of transferred graphene on the SiO₂/Si substrate was around 4,500–6,500 cm²/Vs at 1.4 K (Nature Nanotechnology 2016, 11, 930). As a comparison, the averaged mobility of transferred graphene obtained in this work is about 5100 cm²/Vs at room temperature and the highest value can be 7600 cm²/Vs, which is higher than that reported in our previous work. For the directly grown sample, the averaged mobility is 9100 cm²/Vs at room temperature. This value is also higher than the recently reported ones of graphene single crystals that directly grown on the sapphire substrates (Nature Materials 2022, 21, 740; Science Advances 2021, 7, eabk0115).

For the curve shown in Fig. 4b, the orange one corresponds to the highest mobility of as grown graphene (~10400 cm²/Vs) and the violet one corresponds to the averaged mobility of transferred graphene (~4400 cm²/Vs). We have added these descriptions in the revised Fig. 4b and corresponding captions.

The mobility is calculated by using the equation of $\mu = \frac{dI}{dVg} \cdot \frac{L}{W} \cdot \frac{1}{C_g V_{ds}}$. W/L defines the channel size, where L is defined by the distance between the two electrodes. Considering the actual channel is larger than the electrode width, we calculate the width of the channel W by using equation W=S/L, where S is the area of the graphene between the two electrodes. We have added this information in the revised Method part.

Following the referee's advice, we have shown the original curve to see the position of the Dirac point in Fig. 4b. We can clearly see that as-grown graphene is slightly doped but the transferred one is heavily doped.

We have updated the above changes and discussions in the revised manuscript.

Original comment (10):

9. Supplementary Figure 4 displays optical images of hBN on SiO₂. Notably, images c-h appear almost identical, with the same spots in the same positions. This does not seem to be caused by artifacts from the lens or equipment, as other images most probably taken with the same setup do not exhibit these spots (Sup. Fig. 4a, b, Fig. 3b of the main text...). The authors should provide an explanation for this intriguing observation.

Our reply:

We really thank the referee for pointing out this fault induced during the image editing process. Since our original optical images are not square, in order to arrange the images neatly with the Power Point software, our general processing process is as follows: i) choose one square image as calibration, copy it into eight images, and arrange them evenly and neatly; ii) select target optical images and crop them into the same-sized square image as in step (i), and move them over the images in step (i); iii) set all target images to the bottom layer of the image and delete the 8 calibration images at the top layer. In the last manuscript, six images were not selected when we removed the calibration images in the last step, thus blocking the real optical images (deliberately misaligned and red box marked in Fig. R7c-h to make them could be seen).

We have updated these figures and set them to grayscale in the revised manuscript.

Fig. R7 Optical images of monolayer single-crystal hBN films on SiO₂ substrate at different positions. The blocked images in c-h are marked with red boxes.

Original comment (11):

10. The XPS spectrum only shows the B and N peaks, and there is no discussion about the results. What is the B:N ratio obtained from XPS? Also, does XPS show any evidence of contamination, such as Cu or residues from the etching solution?

Our reply:

We greatly thank the referee for the suggestion on the description of XPS data. The atomic ratio of B:N obtained from the XPS data is 1.08:1, which is very close to 1:1. For the contamination, we don't see the peak corresponding to Cu (Fig. R8a). The etching solution is (NH₄)₂S₂O₈, we also don't see the peak corresponding to S (Fig. R8b). So, there are no detectable contaminations in our method. We have added these data and discussions in the revised manuscript.

Fig. R8 XPS spectra of hBN samples on SiO₂ substrate. The absence of the peaks corresponding to Cu (a) and S (b) indicates no contamination of Cu or (NH₄)₂S₂O₈ on the hBN samples.

Original comment (12):

11. Si coming from the quartz tube at high temperatures can have an influence on the growth of graphene (see e.g., DOI: 10.1038/s41565-020-0743-0). In the current method, the growth of hBN (graphene) occurs very close to the SiO₂ surface (or to the Al₂O₃). Can contamination from the insulator substrate play a role in the growth of the 2D materials here? Is it completely excluded the possible contamination of the 2D material with material from the substrate?

Our reply:

We thank the referee for raising the concern on the influence of Si coming from the quartz tube and providing us the related reference. In the literature, the authors utilized the Si coming from the quartz tube at high temperature to form Cu/Si alloy and realized the growth of layer controlled single-crystal graphene. There is no evidence showing that the quality of graphene is affected by the Si. In our experiment, we didn't find any signal of Si in the XPS spectrum of hBN samples on sapphire substrates (Fig. R9). Although we cannot completely exclude the possible contamination of the 2D materials with material from the substrate, it is safe to say this effect must be very weak if it exists under our growth conditions.

Fig. R9 XPS spectra of hBN samples on sapphire substrate. The absence of the peaks corresponding to Si indicates no doping of the hBN samples.

Original comment (13):

12. Usually, monolayer hBN on Cu presents a very low contrast, so it is a bit surprising that it is so clearly seen in the image of Fig. 2c. As the image is taken from the SiO₂ side, the contrast should not be enhanced by the oxidation of the Cu. Is there any reason for the hBN being that visible?

Our reply:

We thank the referee for this concern on the optical contrast of hBN islands. The referee is totally right that monolayer hBN on Cu presents a very low contrast. For these optical images of hBN taken under normal conditions are typically invisible. In our experiment, we adjusted the brightness, contrast and colour code in the software of the image to identify the hBN islands (the natural colour of the Cu is like that shown in Supplementary Fig. 6c in the last version). We have updated the figure captions in the revised manuscript to avoid this confusion.

Original comment (14):

13. Small mistakes: “height profile” instead of “height distribution” in caption of Fig. 3; “garphene” in caption of Sup. Fig. 6.

Our reply:

We greatly thank the referee for pointing out these typos in the last version and have updated them in the revised manuscript.

In summary, we are very grateful to the referee's efforts in reviewing our manuscript. Especially these valuable comments and suggestions really help us to improve the manuscript significantly. We hope that our reply has fully addressed all raised concerns and the referee will enjoy the story in the revised version.

Reply to Referee #2

Original comment (1):

The manuscript by Zeng et al. reports a stamp-like process that can extrude hBN or graphene from metallic substrates and tightly stick on various insulating substrate. This method demonstrates an “in-situ transfer” process that can transfer hBN or graphene that were grown initially on a Cu foil without using any postgrowth transfer process. Characterizations show that the hBN and graphene are flat and clean with rather good electrical and dielectric performance. The method demonstrated here is very unique and potentially useful for solving the long-standing problems caused by the inevitable postgrowth transfer processes. Therefore, I tend to support publication of this work, after addressing my comments:

Our reply:

We greatly appreciate the referee’s positive evaluation and also his/her valuable suggestions on this work. We have added more experimental data and discussions to address all unclear issues in the last version. We hope the referee will find that the revised version of the manuscript is now suitable for being accepted for publication.

Original comment (2):

1. The title of “stamped growth” is quite confusing. This method is actually a stamp-like transfer process after growth, not exactly a growing process. I suggest the authors to change this term.

Our reply:

We thank the referee for this concern. We agree with the referee that our work is not exactly a traditional growing process. Following the referee’s advice, we have changed the title as “Stamped production of single-crystal hexagonal boron nitride monolayers on various insulating substrates” in the revised manuscript, where the word “production” contains both the growth and transfer process in one step. And in the text part, we described our idea more clearly about the “one-step growth and transfer” design.

Original comment (3):

2. In Fig. 1d-f, how did the authors take the cross-sectional optical images? I am curious about how to take such image at 1080 and 1087 °C. Please explain.

Our reply:

We thank the referee for raising this concern on the cross-sectional optical images. In fact, this is not an in-situ characterization, we first annealed the Cu/SiO₂ at 1080 and 1087 °C, and then turned off the system to room temperature. Finally the cross-sectional optical images can be obtained with a portable microscope (Fig. R1). We have updated the figure captions to make the description clearly in the revised manuscript.

Fig. R1 Schematic diagram (a) and the real setup (b) of the camera to obtain the cross-sectional optical images.

Also, to obtain the accurate distance between Cu foils and the substrate after pre-melting, we have carried out cross-sectional high-resolution transmission electron microscopic (HRTEM) characterization. The atomic characterizations show the distance of Cu and SiO₂ is 3.04 Å (Fig. R2), which is consistent with our theoretical calculations. We have updated this data as Fig. 1f in the revised manuscript.

Fig. R2 Cross-sectional HRTEM image of the Cu/SiO₂ structure.

Original comment (4):

3. Does this method work as well by using a two-step process in two separate furnaces? (i.e, growing 2D materials on a metallic substrate in a furnace and performing the stamp-like transfer in another furnace.) In principle, it should work even in two separate furnaces. If not, do the authors have any comment on this?

Our reply:

We really appreciate the referee for raising this important question. As the referee kindly points out, this technique should in-principle work in two separate furnaces. However, we found that if the graphene samples were taken out of the tube furnace, exposed to air, and then heated to high temperature, the graphene quality was degraded during this process (Fig. R3), possibly due to the contamination and oxidization during the reloading and heating process. Therefore, in our current design, the growth and transfer processes were carried out in one step to avoid additional damage to graphene samples during the extra reloading and heating processes.

Fig. R3 Optical images of as-grown graphene samples (a) and that after annealed in the tube furnace (b) on Cu foils.

Original comment (5):

4. The stamp-like process has to be performed at the melting temperature of Cu. This is a very high temperature, which may damage the substrate with a very thin dielectric layer. Such a high temperature is also not suitable for back-end-of-line processes. Could the authors comment on this?

Our reply:

We are very grateful to the referee for the concern on the influence of the high temperature. In fact, our temperature is only 87 °C higher than the normal growth temperature (about 1000 °C) and the holding time is only 4-8 min. Considering that the melting point of common heat-resisting substrates is about 2000 °C, we believe that this extra temperature increasing with 4-8 min will not bring too much impact compared with traditional growth method. We totally agree with the referee that the high-temperature growth is not suitable for back-end-of-line processes, and this is also a common problem in the entire field of single-crystal hBN growth, and need more innovative explorations. It may be possible to try to grow it on low-melting metals in the future, but it's beyond the subject of this work.

In summary, we are very grateful for all these insightful comments and suggestions from the referee, which helped us to improve the manuscript significantly. We hope that our reply has fully addressed all raised concerns and the referee will enjoy the story in the revised version.

Reply to Referee #3

Original comment (1):

The manuscript entitled "Stamped Growth of Single-Crystal Hexagonal Boron Nitride Monolayers on Various Insulating Substrates" introduces a method of interface growth between Cu foil and insulating substrates. The authors claim that the Cu foil and insulator can be joined together via a premelt process at around the melting point of the Cu foil. After the premelt process, they grew a single-crystal hBN film between the Cu foil and SiO₂, resulting in hBN/Cu/hBN/insulating substrates. The upper hBN layer was etched using plasma treatment, and then the Cu/hBN/insulating substrates were immersed in an (NH₄)₂S₂O₈ solution to remove the Cu, resulting in hBN/insulating substrates. However, the method and results presented in this manuscript lack novelty compared to previous reported research. Additionally, there are several sections of the manuscript that could lead to confusion or misunderstanding. Therefore, this manuscript should be rejected from Nature Communications. I left some representative comments as below.

Our reply:

We greatly thank the referee's time and efforts in reviewing our manuscript, and also his/her valuable suggestions on this work. Maybe we did not show our experimental design clearly, which leads the referee feel that this work is similar to the previous one (Nature Materials 2022, 21, 740; the experimental design is shown in Fig. R1a). In the following, we'd like to explain the main difference and novel advance in our work.

Considering that the growth at the interface is very different and complex, we grew hBN samples with traditional method at 1080 °C (Fig. R1b), the distance between hBN and the SiO₂ substrate at this temperature is about 20 μm, which is wide enough for the diffusion of the boron and nitrogen precursor and the growth of hBN samples on the back surface of Cu foils (Nature Nanotechnology 2016, 11, 930; Nature Chemistry 2019, 11, 730; Scientific Reports 2018, 8, 4046; Chemistry Of Materials 2016, 28, 4893). Then, we increase the temperature of the system, the Cu foils grown with hBN films can stick tightly (at atomic scale) to the insulating substrate at sub-melting temperature of Cu and extrudes the hBN grown on the metallic surface onto the insulating substrate. Single-crystal hBN films can then be obtained by removing the Cu foil similar to the stamp process, regardless of the type or crystallinity of the insulating substrates. This design is not

mentioned in all previous literatures.

The previous work (Nature Materials 2022, 21, 740) is very important and first realized the growth of single-crystal graphene films on c-plane sapphire substrates. Compared to that work, the advantage of our designs includes the following points. (i) hBN grows directly on the back surface of the Cu foil, rather than at the interface by the very slow diffusion. (ii) The only requirement for the substrate is the heat resistance, rather than the previous single-crystal c-plane sapphire. (iii) The single crystallinity and quality of the prepared hBN samples should be almost the same as those grown on Cu. (iv) Not only graphene but also hBN single crystals on various insulating substrates can be produced in our method.

We have updated the manuscript to describe the experimental design clearly, we hope the referee could agree that our design is explicitly different to previous works.

Fig. R1 Schematic diagrams of the growth process of graphene/hBN on insulating substrates in the reference (a) and our manuscript (b). The growth process is conducted after the adhesion of Cu and Al₂O₃ in the reference. In our work, the growth process is conducted before the adhesion of Cu and SiO₂.

Original comment (2):

1. Many previous studies have achieved monolayer single-crystalline hBN films. In this work, the authors only applied interface growth, and although they demonstrated a chemical etching

process to remove the upper hBN and Cu, they also transferred the prepared hBN/SiO₂ to another substrate. Therefore, the novelty of this work compared to previous reports is unclear.

Our reply:

We totally agree with the referee that several previous studies have achieved monolayer single-crystal hBN films. However, we respectfully point out that all these hBN films are obtained on metal surface like Cu and Au (Science 2018, 362, 817; Nature 2019, 570, 91; Nature 2020, 579, 219), and the growth of monolayer single-crystal hBN on insulating substrates has not been achieved so far. To match mature industrial technology, the preparation of single-crystal hBN films on insulating substrates without transfer is very important, as the transfer process would inevitably introduce uncontrollable contamination and mechanical damage, which would greatly degrade the performance of the devices (Nature Communications 2019, 10, 1912; Nature Communications 8, 14560; Science Advances 2015, 1, e1500222).

As for transferring hBN/SiO₂ to the Au coated SiO₂/Si substrates in our manuscript, the purpose is to directly measure the dielectric constant of hBN following the editor's suggestion, as these characterizations require that the hBN should be placed on a conducting substrate (Nature 2020, 582, 511; Adv. Funct. Mater. 2018, 1804235). However, for real applications, this additional transfer process will no longer be required and the hBN/SiO₂ can be used directly.

We hope the referee will agree that we have first realized the production of monolayer single-crystal hBN films on various insulating substrates. We have updated these discussions and emphasized our novelty in the revised manuscript.

Original comment (3):

2. The concept of this work is similar to a previous report [Nature Communications 11, 849 (2020)]. In that report, the metal substrate could be easily removed after growth. In contrast, this work applied chemical etching and transfer methods.

Our reply:

We greatly thank the referee very much for providing us this related reference. In this mentioned work, the boron and nitrogen precursor first dissolved into the Fe₈₂B₁₈ alloy and then segregated on the interface between the sapphire and Fe-B alloy. However, that design cannot

control the layer number and lattice orientations of the segregated islands, and the growth of monolayer single-crystal films was not realized.

Therefore, the current production of single-crystal hBN films can only be achieved on a metal substrate with a self-limiting growth mode. Our design adopts this strategy and realize the growth and transfer simultaneously. Except for chemical etching, the Cu foils can also be peeled off directly and left millimetre-scale monolayer single-crystal graphene films (Fig. R2). Even if we used the chemical etching method, no polymer was used in this process and the final hBN film can be quite clean. We believe that the clean hBN on insulating substrate will be critical for future industrial production.

We have added these discussions in the revised manuscript.

Fig. R2 a, Schematic diagram of the production of single-crystal graphene without chemical etching. b-c, Optical images of graphene samples produced by peeling off the Cu foil.

Original comment (4):

3. If this work can demonstrate interface growth without the need for transfer processes and is applicable to universal substrates such as PET film or glass, it would have strong novelty.

Our reply:

We totally agree with the referee that it will be quite important if one could realize the growth of single-crystal hBN (graphene) on various insulating substrates, containing low-melting substrates like PET film or glass. Indeed our method itself has no limitation to grow on any insulating substrate. The only limitation come from the growth temperature. In the future, if one can realize the single-crystal hBN (graphene) growth on low-melting substrates, our method will be directly adopted and extended to various substrates. We think there will be continuous studies in the field on this

important direction.

Original comment (5):

4. Finally, the manuscript lacks clear explanations about the preparation of samples, measurements, and analysis. Therefore, all specific details should be included in the manuscript.

Our reply:

We thank the referee for this kind suggestion. In the last version of the manuscript, we showed the production of samples, measurements and analysis in the Method part. Following the referee's advice, we have updated the Method part in details to provide clearer explanations.

In summary, we are very grateful for all these tough comments and critical suggestions from the referee, which helped us to improve the manuscript significantly. We hope that our reply has fully addressed all raised concerns and the referee will enjoy the story in the revised version.

REVIEWER COMMENTS

Reviewer #1 (Remarks to the Author):

I would like to thank the authors for considering my previous comments, and for their effort with the explanations and the new added experiments. Overall, I think that they satisfactorily addressed most of my concerns. There are still two minor things mentioned below that I would appreciate if the authors could consider before finalizing the manuscript. Other than that, I think that the results are quite interesting, and I can recommend the publication of the manuscript.

Concerning the reply to original comment #8, I still think that the choice to present the data is not that correct.

The meaning of “traditional” in the supplementary information is still not mentioned, which could lead to misinterpretations. The included comparison with direct growth on insulating substrates (coined as “traditional” here) is necessary, but the authors also need to show that the current method works at least as good as the “traditional” PMMA transfer.

Otherwise, I am sorry to say that I still think that the way this data is presented (misleading “traditional” term plus no comparison with which is usually referred to as “traditional”) is biased. I would suggest to the authors to properly indicate in the SI what do they mean by traditional (i.e., established/existing methods to grow graphene on dielectric substrates) and/or to include data from transferred graphene to avoid any confusion or misinterpretation.

About the reply to original comment #9, I still think that the choice of devices in fig. 4c is biased and not proper. In fact, this is reinforced by the information given in the reply.

First, the device chosen to represent the method proposed in the manuscript is the one with the highest mobility value, whereas for the transferred graphene a device with an “average” mobility of 4400 cm²/Vs was chosen. This difference of criterium alone makes the comparison unfair.

Second, the value of the average of 5100 cm²/Vs probably includes the outlier device number 2: if this device is removed, a device with a mobility of 4400 cm²/Vs is in fact one of the worst performing devices. Hence, the comparison turns out to be between the best

device of one method against one of the worst for the other method. This makes such comparison much more unfair than it was already.

Finally, the chosen device for transferred graphene shows a quite broad conductance minimum, which actually seems to have a secondary component close to 0 V. As mobility is calculated from the slope of the curve, the presence of this component obviously leads to lower mobility values. This might indicate a problem with that device, but it is not clear if this is the case for all the transferred devices or this device is a special one.

Reviewer #2 (Remarks to the Author):

After reading all the comments/replies, I found the revised manuscript has been improved considerably. All of the questions I raised have been addressed properly. I also noted that the other two reviewers have raised many valuable questions and comments, especially on the novelty of this work. In my opinion, the in-situ growth and transfer of hBN on various insulating substrate is novel and useful. Although, as I mentioned, high temperature is a concern, currently the method reported here is still a good method to obtain clean hBN on insulating substrate. I believe that the revised manuscript with added new data is justified for publication in Nature Communications.

Reviewer #3 (Remarks to the Author):

The authors revised some figures; however, the exhibited method and results still show lack novelty compared to previous reported research. The main novelty of this work looks the production of single crystal hBN and graphene films on the various insulating substrates without transfer process. But the results are not clear to prove the authors' opinions. Therefore, this manuscript should be major revision to be more clarify the authors' thesis. The representative reasons are as follows.

1. An important aspect that should be addressed is the absence of representation of the hBN film grown on the upper side of Cu in Figure 1a. This omission may result in a misunderstanding of the growth mechanism. It can be a minor problem. However, there are some parts that lead the misunderstanding in this manuscript. I cannot point out all things.

Please double-check all data that can lead the misunderstanding.

2. In figure 1c, without the dot line, we can not see the grown hBN flakes. Like this, the most figures about the morphologies of the grown hBN and graphene are indistinguishable between the hBN and the substrate (figure 3b, supplementary figures 5, 9a-c, 10c,e, 11a-c). Thus, they should be all replaced to the distinguishable images.

3. As the substrate for growth, the authors have used the SiO₂(001), SrTiO₃(001), c-plane sapphire, fused silica. The shown optical image of an as-grown hBN sample shows only white circle substrate (figure 3a). Likewise, we cannot distinguish the grown hBN in the optical image. Please add more growth results on other normal substrates such as a SiO₂ (300nm)/Si substrate.

4. To demonstrate the quality of the single crystal hBN and graphene films in large-scale, the authors should update the statistics data, such as FWHM and peak position, which are extracted from Raman mapping images. They will be more effective data to confirm the quality of large-scale single crystal hBN and graphene film.

5. The explanation provided to prove the single crystallinity of the film through Supplementary Figure 7 is inadequate. There is a lack of evidence supporting the presence of single crystallinity in both the etched and unetched areas. Considering the growth of graphene and hBN on the Cu (111) surface via hydrogen etching at high temperatures, it is important to note that the etching process primarily targets the energetically unstable areas, rendering it unsuitable as evidence for single crystals. Thus, the TEM analysis in large-scale is enough to confirm the single crystallinity of the grown hBN film.

Reply to Referee #1

Original comment (1):

I would like to thank the authors for considering my previous comments, and for their effort with the explanations and the new added experiments. Overall, I think that they satisfactorily addressed most of my concerns. There are still two minor things mentioned below that I would appreciate if the authors could consider before finalizing the manuscript. Other than that, I think that the results are quite interesting, and I can recommend the publication of the manuscript.

Our reply:

We greatly thank the referee for his/her time and expertise in reviewing our manuscript. We also thank his/her very helpful comments for us to further improve the quality of this work. We have updated the choice of control samples in Raman and FET devices to address the referee's concerns as in the below replies.

Original comment (2):

Concerning the reply to original comment #8, I still think that the choice to present the data is not that correct.

The meaning of “traditional” in the supplementary information is still not mentioned, which could lead to misinterpretations. The included comparison with direct growth on insulating substrates (coined as “traditional” here) is necessary, but the authors also need to show that the current method works at least as good as the “traditional” PMMA transfer.

Otherwise, I am sorry to say that I still think that the way this data is presented (misleading “traditional” term plus no comparison with which is usually referred to as “traditional”) is biased. I would suggest to the authors to properly indicate in the SI what do they mean by traditional (i.e., established/existing methods to grow graphene on dielectric substrates) and/or to include data from transferred graphene to avoid any confusion or misinterpretation.

Our reply:

We greatly thank the referee for raising the concern on the possible confusion induce by the word “traditional”. As the referee kindly pointed out that the traditional method in our manuscript

not corresponds to the PMMA transfer method, in the last version of the manuscript, we have clearly stated that the samples for comparison were directly grown on the quartz glass substrates using a catalyst-free method. To make this point clear, we have replaced “traditional method” to “catalyst-free growth method”. Now, Supplementary Fig. 12g and the captions are as follows:

Fig. R1. ID/IG ratio of 40 points of different samples prepared from catalyst-free growth method and stamp method. The green and orange colours correspond to catalyst-free growth method and stamp method, respectively.

We have updated this part in the revised Supplementary information.

Original comment (3):

About the reply to original comment #9, I still think that the choice of devices in fig. 4c is biased and not proper. In fact, this is reinforced by the information given in the reply.

First, the device chosen to represent the method proposed in the manuscript is the one with the highest mobility value, whereas for the transferred graphene a device with an “average” mobility of $4400 \text{ cm}^2/\text{Vs}$ was chosen. This difference of criterium alone makes the comparison unfair.

Second, the value of the average of $5100 \text{ cm}^2/\text{Vs}$ probably includes the outlier device number 2: if this device is removed, a device with a mobility of $4400 \text{ cm}^2/\text{Vs}$ is in fact one of the worst performing devices. Hence, the comparison turns out to be between the best device of one method against one of the worst for the other method. This makes such comparison much more unfair than it was already.

Finally, the chosen device for transferred graphene shows a quite broad conductance minimum, which actually seems to have a secondary component close to 0 V. As mobility is calculated from

the slope of the curve, the presence of this component obviously leads to lower mobility values. This might indicate a problem with that device, but it is not clear if this is the case for all the transferred devices or this device is a special one.

Our reply:

We greatly thank the referee for the suggestion on the comparison of the carrier mobilities. We totally agree with the referee that the choice of the value in the current version may induce inaccurate comparison than it was already. Following the advice, in the revised manuscript, we choose the highest mobility for both the as-grown graphene ($\sim 10400 \text{ cm}^2/\text{Vs}$) and transferred graphene ($\sim 7640 \text{ cm}^2/\text{Vs}$) to make the comparison more reasonable. Now Fig. 4b and the captions are as follows:

Fig. R2. Plot of the conductivity of graphene as a function of gate voltage. The orange and violet curves correspond to the highest mobility of as grown graphene ($\sim 10400 \text{ cm}^2/\text{Vs}$) and transferred graphene ($\sim 7640 \text{ cm}^2/\text{Vs}$), respectively.

For the devices of transferred graphene, we have shown all the curves in Fig. R3. From the curve, we can see that only the curve corresponding to the mobility of $4400 \text{ cm}^2/\text{Vs}$ shows broad conductance minimum (the red curve in Fig. R3), this may be caused by the random defects, doping and scattering by metallic or polymer residues involved by the transfer process.

We have updated the data and discussions in the revised manuscript.

Fig. R3. Plot of the conductivity of transferred graphene as a function of gate voltage.

In summary, we are very grateful for all these insightful comments and suggestions from the referee, which helped us to improve the manuscript significantly. We hope that our reply has fully addressed all raised concerns and the referee will enjoy the story in the revised version.

Reply to Referee #2

Original comment (1):

After reading all the comments/replies, I found the revised manuscript has been improved considerably. All of the questions I raised have been addressed properly. I also noted that the other two reviewers have raised many valuable questions and comments, especially on the novelty of this work. In my opinion, the in-situ growth and transfer of hBN on various insulating substrate is novel and useful. Although, as I mentioned, high temperature is a concern, currently the method reported here is still a good method to obtain clean hBN on insulating substrate. I believe that the revised manuscript with added new data is justified for publication in Nature Communications.

Our reply:

We greatly thank the referee's time and efforts in reviewing this manuscript and his/her recommendation for the publication.

Reply to Referee #3

Original comment (1):

The authors revised some figures; however, the exhibited method and results still show lack novelty compared to previous reported research. The main novelty of this work looks the production of single crystal hBN and graphene films on the various insulating substrates without transfer process. But the results are not clear to prove the authors' opinions. Therefore, this manuscript should be major revision to be more clarify the authors' thesis. The representative reasons are as follows.

Our reply:

We thank the referee's efforts and time in reviewing our manuscript. His/her valuable suggestions and comments are quite helpful for us to improve the quality of this work. We have added more experimental data to address these concerns as in the below replies.

Original comment (2):

1. An important aspect that should be addressed is the absence of representation of the hBN film grown on the upper side of Cu in Figure 1a. This omission may result in a misunderstanding of the growth mechanism. It can be a minor problem. However, there are some parts that lead the misunderstanding in this manuscript. I cannot point out all things. Please double-check all data that can lead the misunderstanding.

Our reply:

We greatly thank the referee for the kind suggestion on the schematic diagrams of our growth process. Following the referee's advice, we have added the hBN grown on the upper surface of Cu in Fig. 2a to make the growth mechanism clearer (Fig. R1). We have also double-checked our manuscript carefully to eliminate possible misunderstandings.

Fig. R1. Schematic diagrams of the growth process. The hBN islands are first grown on the back surface of the Cu foil and then tightly stick to the SiO₂ substrate. Finally, the Cu foil was removed to obtain hBN samples.

Original comment (3):

2. In figure 1c, without the dot line, we cannot see the grown hBN flakes. Like this, the most figures about the morphologies of the grown hBN and graphene are indistinguishable between the hBN and the substrate (figure 3b, supplementary figures 5, 9a-c, 10c, e, 11a-c). Thus, they should be all replaced to the distinguishable images.

Our reply:

We greatly thank the referee for the suggestion on the morphologies of the grown hBN and graphene samples. For hBN in Fig. 2c and graphene in Supplementary Fig. 10c, the image is taken from the SiO₂ side, so the contrast cannot be enhanced by the oxidation of the Cu. As monolayer hBN/graphene on Cu presents a very low contrast, the hBN/graphene flakes are not clear without the dot line. In the revised manuscript, we adjusted the brightness, contrast and colour code in the software of the image to make the hBN/graphene islands visible and added the figures as Supplementary Fig. 4 (Fig. R2a-b).

For continuous hBN/graphene films shown in Fig. 3b, Supplementary fig. 5, 9a-c, 10c, e, and 11a-c), some scratches are deliberately made to make the substrate and hBN/graphene samples distinguishable in the revised manuscript (Fig. R2c is an example of updated figure for Fig. 3b).

We have updated the figures in the revised manuscript and Supplementary information.

Fig. R2. **a-b**, Optical images of hBN and graphene taken from the SiO₂ side. The images are adjusted to make the islands visible. **c**, optical images of hBN films on SiO₂ substrate. The top right corner in (c) is an intentional scratch.

Original comment (4):

3. As the substrate for growth, the authors have used the SiO₂(001), SrTiO₃(001), c-plane sapphire, fused silica. The shown optical image of an as-grown hBN sample shows only white circle substrate (figure 3a). Likewise, we cannot distinguish the grown hBN in the optical image. Please add more growth results on other normal substrates such as a SiO₂ (300 nm)/Si substrate.

Our reply:

We greatly thank the referee for raising the concern on the optical images of hBN samples. Due to the very low contrast of hBN, the hBN films is hard to distinguish directly. Following the referee's advice, we have added the growth results of hBN on SiO₂/Si substrate and the contrast is also very low. To make the samples and substrate distinguishable, some scratches are deliberately made (Fig. R3), and we have added this data in the revised Supplementary figures.

Fig. R3. **a-b**, Optical images of hBN grown on the SiO₂/Si substrate. **c**, Zoom-in image of hBN samples. The images are adjusted to make the islands visible. **d**, optical images of hBN films on

SiO₂ substrate. **d**, Statistical distributions of the E_{2g} -band position and FWHM of hBN.

Original comment (5):

4. To demonstrate the quality of the single crystal hBN and graphene films in large-scale, the authors should update the statistics data, such as FWHM and peak position, which are extracted from Raman mapping images. They will be more effective data to confirm the quality of large-scale single crystal hBN and graphene film.

Our reply:

We greatly thank the referee for the kind suggestion on the statistical Raman data. Following the referee's advice, we have carried out Raman mapping at large scale and extracted the data to show the statistical distributions of the 2D-band position and FWHM of graphene (Fig. R4a for Supplementary Fig.12f), and the E_{2g} -band position and FWHM of hBN (Fig. R4b for Fig.3d). These statistical distributions provide very intuitive data to show the high uniformity of the samples. We have also updated the statistics data in Supplementary Fig. 11e-h and Supplementary Fig. 13d-f in the revised Supplementary information.

Fig. R4. Statistical distributions of the 2D-band position and FWHM of graphene (**a**) and E_{2g} -band position and FWHM of hBN (**b**).

Original comment (6):

5. The explanation provided to prove the single crystallinity of the film through Supplementary

Figure 7 is inadequate. There is a lack of evidence supporting the presence of single crystallinity in both the etched and unetched areas. Considering the growth of graphene and hBN on the Cu (111) surface via hydrogen etching at high temperatures, it is important to note that the etching process primarily targets the energetically unstable areas, rendering it unsuitable as evidence for single crystals. Thus, the TEM analysis in large-scale is enough to confirm the single crystallinity of the grown hBN film.

Our reply:

We greatly thank the referee for the kind suggestion on the TEM characterizations. Following the referee's advice, we have conducted selected area electron diffraction (SAED) and high-resolution transmission electron microscopy (HRTEM) characterizations at different areas of the TEM grids. The identical diffraction patterns in reciprocal space and atomic lattices in real space all demonstrate the single-crystal nature of the grown hBN films (Fig. R5-6).

We have added the TEM data in the revised Supplementary information.

Fig. R5. Typical SAED patterns of as-grown hBN samples. The nearly identical crystallographic orientations reveal the single-crystal structure of the sample.

Fig. R6. Typical HRTEM images of as-grown hBN samples. The nearly identical lattice reveals the single-crystal structure of the sample.

In summary, we are very grateful for all these valuable comments and suggestions from the referee. We hope that our reply has fully addressed all raised questions and the referee will find the revised version has greatly improved.

REVIEWERS' COMMENTS

Reviewer #1 (Remarks to the Author):

The authors have properly addressed the minor issues that I previously raised regarding the manuscript. Given the relevance and potential impact of the work, allowing to produce high-quality hBN and graphene on insulating substrates, I consider that it merits publication in Nature Communications.

Reviewer #3 (Remarks to the Author):

The authors have appropriately addressed my comments and have also updated the relevant data. Therefore, I believe that the completeness of this paper has improved, making it suitable for acceptance. Therefore, I recommend that this manuscript be accepted for publication in Nature Communications journal without further revisions.